# Hepatocellular EVs Regulate Lipid Metabolism via SIRT1/SREBP−1c/PGC−1α Signaling in Primary Calf Hepatocytes

**DOI:** 10.3390/ijms26199392

**Published:** 2025-09-25

**Authors:** Daoliang Zhang, Jishun Tang, Leihong Liu, Chang Zhao, Shibin Feng, Xichun Wang, Hongyan Ding, Yu Li

**Affiliations:** 1College of Veterinary Medicine, Anhui Agricultural University, Hefei 230061, China; zdl15936569251@163.com (D.Z.); liuleihong100@163.com (L.L.); chang_zhao@ahau.edu.cn (C.Z.); fengshibin@ahau.edu.cn (S.F.); wangxichun@ahau.edu.cn (X.W.); 2Hefei Institutes of Physical Science, Chinese Academy of Sciences, Hefei 230031, China; 3Anhui Province Key Laboratory of Livestock and Poultry Product Safety, Institute of Animal Science and Veterinary Medicine, Anhui Academy of Agricultural Sciences, Hefei 230001, China; tjs157@163.com

**Keywords:** calf hepatocytes, ketosis, extracellular vesicles, lipid metabolism disorder, SIRT1/SREBP−1c/PGC−1α signaling pathway

## Abstract

SIRT1-SREBP−1c/PGC−1α signaling is involved in the production of non-esterified fatty acids (NEFAs) and liver lipid metabolism disorders in ketotic calf. The molecules contained in extracellular vesicles (EVs) regulate intercellular communication, and research on calf hepatocytes−derived EVs has become a hot spot. We hypothesized that EVs in cell culture supernatants could affect lipid metabolism in hepatocyte models via SIRT1/SREBP−1c/PGC−1α signaling. Non-ketosis (NK, 0 mM NEFA) and clinical ketosis calf models (CK, 2.4 mM NEFAs) were established in vitro cultured calf hepatocytes and EVs were extracted from their supernatants as NK−derived EVs and CK−derived EVs, respectively. In vitro hepatocyte models, comprising a normal culture group (normal) and the group treated with NEFAs at 2.4 mM (2.4 NEFA), were treated with NK and CK−derived EVs. In addition, we transfected an SIRT1−overexpressing adenovirus into calf hepatocytes and determined the expression of key genes, enzymes, and proteins involved in the SIRT1/SREBP−1c/PGC−1α pathway. The results showed that the NK−derived EVs inhibited the expression of the SREBP−1c gene and protein and increased the expression of the SIRT1 and PGC−1α genes and proteins (*p* < 0.05). In contrast, CK−derived EVs induced lipid metabolism disorders in the normal hepatocyte group and aggravated NEFA-induced lipid metabolism imbalances in hepatocytes (*p* < 0.05). Moreover, overexpression of SIRT1 confirmed that EVs exert vital functions in hepatocyte lipid metabolism via SIRT1/SREBP−1c/PGC−1α signaling to regulate hepatocyte lipid metabolism. In summary, NK−derived EVs alleviated liver lipid metabolism disorders caused by NEFAs via modulation of SIRT1/SREBP−1c/PGC−1α signaling, while CK−derived EVs had the opposite effect. NK−derived EVs upregulated lipid oxidation-related genes and downregulated lipid synthesis-related genes, suggesting that NK−derived EVs could be used as biological extracts to alleviate lipid metabolism disorders in ketotic calf.

## 1. Introduction

The decrease in dry matter intake of calf and the increase in energy demand during lactation, which is manifested as a negative energy balance (NEB), make them prone to metabolic diseases such as ketosis, thereby increasing the economic losses of the calf industry [1,2]. Calf ketosis is a nutritional metabolic disease involving liver lipid metabolism disorders and is characterized by high levels of non-esterified fatty acids (NEFA). NEB initiates fat mobilization, resulting in an increase in the concentration of blood of NEFAs in calf. The liver is the main location of NEFA metabolism. When the intake of NEFAs exceeds the esterification or oxidation capacity of the liver, excessive triglyceride (TG) accumulation leads to liver lipid deposition or fatty liver [3]. Recently, extracellular vesicles (EVs) have become a research hotspot and there are many reports on the regulation of physiological functions by EVs. Consequently, this study aimed to explore whether cultured cell−derived EVs can modulate hepatic lipid metabolism.

Extracellular vesicles, a type of exosome comprising double-membrane vesicles with a diameter of 50–150 nm, contain microRNAs, mRNAs, lipids, and proteins derived from their from parental cells, and are secreted actively into the extracellular milieu by all types of cells [4,5]. Target cells take up and internalize EVs via receptors, endocytosis, or membrane fusion to achieve intercellular communication. Studies have reported a relationship between circulating EVs and hepatic metabolism [6,7]. Exosomes can transmit information from donor cells to target cells, thus regulating the metabolism, proliferation, apoptosis, and other functions of liver cells. For example, under lipotoxic conditions, calf hepatocytes−derived exosomes can inhibit peroxisome proliferator-activated receptor (PPAR) γ -mediated activation of hepatic stellate cells through vanin−1 and encapsulated miR−128−3p on their surface, leading to the induction of pro-fibrotic gene expression [8]. Calf hepatocytes release hepatocyte−derived extracellular vesicles (hep-EVs) in response to injury or stress, and changes in their composition and number have been associated with the progression of non-alcoholic fatty liver disease, non-alcoholic hepatitis, and alcoholic liver disease. Detection of hep-EVs in the blood or tissues provides an insight into the severity and progression of liver disease [9]. Wu’s group, through miR-130a-3p knockout and overexpression, determined that miR−130a-3p in hepatic EVs mediated crosstalk between the liver and adipose tissue, and regulated the energy metabolism of adipose tissue [10]. The transfer of miR−199a-5p from EVs into mice regulated liver lipid accumulation by regulating liver sterol regulatory element binding transcription factor 1 (SREBP-1c), protein kinase AMP-activated catalytic subunit alpha 1 (AMPKα), carnitine palmitoyltransferase 1alpha, and fatty acid synthase (FAS) [11]. Chen et al. observed that labeled EVs were transported from the intestine to the liver, triggering hepatic steatosis [12]. Herein, we revealed the mechanism by which in vitro cultured cell model−derived EVs regulate lipid synthesis and transport in calf hepatocytes through the sirtuin 1 (SIRT1)/SREBP−1c/peroxisome proliferator-activated receptor gamma coactivator 1−alpha (PGC−1α) signaling pathway.

In calf, liver lipid metabolism is regulated by the SIRT1/SREBP−1c/PGC−1α signaling pathway via one or more factors. Studies have already shown that in vitro, NEFA treatment increased the expression of SREBP−1c and activated the nuclear transcription factor-κB (NF-κB) pathway. Moreover, SREBP−1c increased the NEFA-induced excessive activation of NF-κB inflammatory pathway by increasing hepatocyte reactive oxygen species (ROS) in calf and promoted the synthesis of TG and inflammatory cytokines, thereby further increasing liver inflammatory injury/fatty liver [13]. Wang’s group found that low concentrations of NEFA promoted AMPKα phosphorylation in calf hepatocytes cultured in vitro. Phosphorylation of AMPKα negatively regulates the expression and transcriptional activity of SREBP−1c and activates SIRT1, which deacetylates the downstream target gene PGC1α [14,15]. This results in the inactivation of lipid synthesis genes such as encoding acetyl-CoA carboxylase alpha and FAS, thereby regulating fat formation [16]. The AMPKα signaling pathway is involved in lipid metabolism regulation in calf hepatocytes. Activation of AMPKα will increase the levels of SIRT1, PGCα, PPARα, and other proteins that promote lipid transport, inhibit or reduce SREBP−1c and acyl-coa oxidase (ACO) to promote lipid oxidation and transport, inhibit lipid synthesis, and participates in adaptive regulation of energy metabolism [17]. In summary, EVs, as potential biomarkers to identify the status of liver function of calf, are also key participants in the body’s energy homeostasis. They play an important role in the lipid and total cholesterol metabolism regulated by the SIRT1/SREBP−1c/PGC−1α signaling pathway in calf hepatocytes, which is worthy of further study.

Based on the hypothesis that EVs may regulate intercellular communication during hepatic lipid metabolism disorders, we investigated the role of calf hepatocytes−derived EVs in the hepatocyte ketosis model in cows.

## 2. Results

### 2.1. Characterization of Hepatic Cells and EVs

As shown in Figure 1b–e, the hepatocyte surface marker CK18 was expressed on the cultured calf hepatocytes. To confirm the preparation and characterization of the prepared enriched EVs, we used Western blotting, TEM, and NTA methods. Western blotting confirmed the expression of transmembrane proteins CD9, CD63, and HSP70 on the surface of the EVs and in the cell culture supernatant. The enriched exosome suspensions from NK and CK calf hepatocytes showed that the diameters of the EVs peaked between 100 nm and 150 nm. The EVs derived from calf hepatocytes stained by phosphotungstic acid were saucer-shaped vesicles with a very obvious membrane structure at 50–150 nm under TEM, and the EVs obtained by ultracentrifugation were highly enriched. There was no significant difference in the exosome appearance between the NK and CK sources.

### 2.2. Determination of the Optimum Concentration and Treatment Time of EVs

The extracted EVs were applied to hepatocytes treated with 0 or 2.4 mM NEFAs (non-ketosis model group and clinical ketosis model group) (Figure 1a). Overlapping images showed that the EVs could be internalized into liver cells, in which they aggregated in the cytoplasm. Calf hepatocytes reached their optimum state after adherent culture for 48 h. Gradient concentrations of enriched NK and CK EVs (0, 20, 40, 60 and 80 μg/mL) were added to the cultured calf hepatocytes. After 12 h and 24 h with or without high NEFA treatment, the viability of the calf hepatocytes was detected using the CCK-8 method. The results showed that for the normal and 2.4 mM NEFA-treated calf hepatocytes, EVs at 40 μg/mL was the exosome concentration and the optimal treatment time was 12 h (Figure 1f). The green fluorescent dye PKH67 was used to detect exosome internalization by calf hepatocytes. Exomes with surface PKH67 labeling were co-cultured with calf hepatocytes (Figure 1g).

### 2.3. Effects of NK−Derived EVs on SIRT1/SREBP−1c/PGC−1α Pathway-Related Gene and Protein Expression in Normal Calf Hepatocytes and Those Treated with NEFAs at 2.4 mM

Herein, qRT-PCR was used to detect the effects of NK EVs on the relative mRNA expression of SIRT1/SREBP−1c/PGC−1α pathway-related genes in normal calf hepatocytes (normal group) and 2.4 mM NEFA-treated calf hepatocytes (NEFA group). As shown in Figure 2a, the NK−derived EVs had no significant effect on normally cultured calf hepatocytes. When the exosome concentration was 40 μg/mL, the NK-derived EVs played a more significant role in regulating the relative mRNA expression of SIRT1/SREBP-1c/PGC−1α pathway-related genes in the 2.4 mM NEFA group (NK−2.4NEFA) than in the normal group (NK-Normal). Treatment with 40 μg/mL NK EVs showed the most significant effect and the relative mRNA levels of SREBP1C and FAS, which are genes related to lipid synthesis, were extremely significantly reduced (*p* < 0.01). The relative mRNA expression levels of lipolysis-related genes SIRT1, AMPKA, and ACO were significantly increased (*p* < 0.01) and the relative mRNA expression of PGC1α was significantly increased (*p* < 0.05). The results suggested that NK−derived EVs could alleviate the disordered lipid metabolism of 2.4 mM NEFA-treated calf hepatocytes via the SIRT1/SREBP1c/PGC1α pathway, and also regulated genes related to fat synthesis and lipoxygenase in calf hepatocytes.

The impacts of NK−derived EVs on the levels of SIRT1/SREBP−1c/PGC−1α pathway-related proteins in the normal and the NEFA groups were detected using Western blotting and immunofluorescence. Figure 2b,c shows the gray-scale scans of SIRT1/SREBP−1c/PGC−1α pathway protein levels in the normal group treated with NK exosomes and in the NEFA group. As shown in Figure 2d, compared with treatment with EVs at 0 μg/mL, the treatment of normal calf hepatocytes with different concentrations of NK-derived EVs resulted in significant changes in the levels of SIRT1/SREBP-1c/PGC−1α pathway-related proteins. This suggested that NK−derived EVs could regulate liver lipid metabolism−related proteins in normal calf hepatocytes via the SIRT1/SREBP−1c/PGC−1α pathway. As shown in Figure 2e, compared with those not treated with EVs, calf hepatocytes treated with NEFAs at 2.4 mM and then with different concentrations of NK−derived EVs showed changes in the levels of SIRT1/SREBP−1c/PGC−1α pathway-related proteins consistent with attenuated lipid metabolism disorders. Moreover, when the NEFA group was treated with 40 μg/mL NK−derived EVs, the levels of SIRT1, AMPKα, PGC−1α, and ACO were significantly increased (*p* < 0.01) and the levels of pAMPKα/AMPKα, SREBP−1c, and FAS were significantly decreased (*p* < 0.01), all of which were consistent with the results obtained using qRT−PCR.

### 2.4. Effects of NK EVs on SREBP−1c and PGC−1α Expression and SIRT1/SREGBP−1c/PGC−1α Pathway-Related Enzyme Activity in Normal and 2.4 mM NEFA-Treated Dairy Cow Hepatocytes

Figure 3a,b shows the immunofluorescence detection of SREBP−1c and PGC−1α proteins in the NK-derived EV−treated normal group and the NEFA group. Herein, the impacts of NK EVs on SIRT1/SREBP−1c/PGC−1α pathway-associated enzyme activities in calf hepatocytes treated by NEFAs and in the normal control group were examined using ELISA. As shown in Figure 3c−f, compared with the control group treated with EVs at 0 μg/mL, the enzyme activities of SIRT1 and pAMPKα were significantly increased in normal hepatocytes treated with NK EVs at 40 μg/mL (*p* < 0.05), but had no significant effects on other proteins. When the 2.4 mM NEFAs were used to treat calf hepatocytes, compared with the control group, the enzyme activities of SIRT1 and pAMPKα were very significantly increased (*p* < 0.01), that of PGC−1α was significantly increased (*p* < 0.05), and SREBP−1c activity was significantly decreased (*p* < 0.05) when 40 μg/mL NK−derived EVs were used. The above results indicated that NK EVs affected the activities of related enzymes through the SIRT1/SREBP−1c/PGC−1α pathway to alleviate the lipid metabolism disorder of 2.4. mM NEFA-treated hepatocytes. Treatment with 40 μg/mL NK EVs had the most pronounced effect on the enzyme activities related to lipogenesis and lipoxidation pathways in hepatocytes.

### 2.5. Effects of CK EVs on the Relative mRNA and Protein Expression Levels of Genes Associated with the SIRT1/SREBP−1c/PGC−1α Pathway in Normal and 2.4 mM NEFA−Treated Dairy Cow Hepatocytes

We used qRT−PCR to detect the effects of CK EVs on the relative mRNA expression levels of SIRT1/SREBP−1c/PGC−1α pathway-related genes in normal calf hepatocytes (Normal group) and 2.4 mM NEFA−treated calf hepatocytes (NEFA group). In the normal group, CK−derived exosomes significantly regulated the relative mRNA expression levels of genes associated with the SIRT1/SREBP−1c/PGC−1α pathway compared with those in the 0 μg/mL group (Figure 4a). These results suggested that CK exosomes might induce lipid metabolism disorders in normal calf hepatocytes via the SIRT1/SREBP−1c-PGC−1α pathway. CK−derived exosomes acting on 2.4 mM NEFA-treated cow hepatocytes significantly altered the relative expression levels of genes associated with the SIRT1/SREBP−1c/PGC−1α pathway in the 40 μg/mL EV treatment group compared with those in the 0 μg/mL group. These findings suggested that CK−derived EVs might exacerbate lipid metabolism disorders in NEFA-treated hepatocytes via SIRT1/SREBP−1c/PGC−1α signaling.

The effects of CK−derived EVs on SIRT1/SREBP−1c/PGC−1α pathway protein levels in the normal group and the NEFA group were detected by Western blotting and immunofluorescence. The gray-scale scan results for the protein levels in the SIRT1/SREBP−1c/PGC−1α pathway in the CK EV-treated normal group and the NEFA group are shown in Figure 4b,c. The results showed that CK−derived EVs could promote the hepatic lipid metabolism disorder of normal hepatocytes via SIRT1/SREBP−1c/PGC−1α signaling (Figure 4d). As shown in Figure 4e, CK EVs at 40 μg/mL could aggravate the lipid metabolism disorder of hepatocytes through the SIRT1/SREBP−1c/PGC−1α pathway in the 2.4 mM NEFA-treated calf hepatocytes, in which the protein levels of PGC−1α and ACO decreased extremely significantly (*p* < 0.01), the protein level of AMPKα increased significantly (*p* < 0.05) and the protein levels of SREBP−1c and FAS increased extremely significantly (*p* < 0.01).

### 2.6. Effects of CK EVs on SREBP−1c and PGC−1α Expression and SIRT1/SREGBP−1c/PGC−1α Pathway−Related Enzyme Activity in Normal and 2.4 mM NEFA-Treated Dairy Cow Hepatocytes

Figure 5a,b shows the immunofluorescence results for SREBP−1c and PGC−1α in the CK−derived exosome-treated normal group and the NEFA group. The effects of CK−derived EVs on SIRT1/SREBP-1c/PGC−1α pathway-associated enzyme activities in calf hepatocytes treated by NEFA and in the normal control group were detected using ELISA. As shown in Figure 5c–f, the results suggested that treatment of normal calf hepatocytes with different concentrations of CK-derived EVs could increase lipogenesis in hepatocytes via SIRT1/SREBP−1c/PGC−1α signaling, reduced the activities of enzymes related to the lipoxidation pathway, and increased the risk of liver lipid accumulation. In the bovine hepatocyte model treated with high NEFA (2.4 mM), treatment with 40 μg/mL CK−derived EVs significantly reduced PGC−1α enzymatic activity (*p* < 0.01), decreased SIRT1 expression (*p* < 0.05), and markedly upregulated SREBP−1c levels (*p* < 0.01) compared with the group not treated with CK EVs. These results indicated that 40 μg/mL CK EVs had the most significant effect on SIRT1/SREBP-1c/PGC−1α pathway-associated enzyme activities in 2.4 mM NEFA-treated calf hepatocytes.

### 2.7. Effects of SIRT1 Overexpression and Calf Hepatocyte EVs on SIRT1/SREBP−1c/PGC−1α Pathway mRNA and Protein Expression in Normal Hepatocytes and Those Treated with NEFAs at 2.4 mM

In this study, qRT−PCR was used to detect the effects of the blank group (BG), the empty carrier group (EG), the SIRT1-overexpression group (SG), the EG plus 40 μg/mL NK−derived EVs (NK40), (EG + NK40) and SG + NK40; EG plus 40 μg/mL CK−derived EVs (CK40), (EG + CK40), and SG + CK40 on the relative mRNA expression levels of genes related to the SIRT1/SREBP−1c/PGC−1α pathway in normal calf hepatocytes and in those treated with NEFAs at 2.4 mM. The results are shown in Figure 6a. The overexpression of SIRT1 in normal cells affected the SIRT1/SREBP−1c/PGC−1α pathway gene expression, exerting a synergistic effect with NK40 to repair the damage to hepatocytes induced by CK40. For the 2.4 mM NEFA-treated hepatocytes, the overexpression of SIRT1 and NK40 treatment both relieved the hepatic lipid metabolism disorder through the SIRT1/SREBP−1c/PGC−1α pathway and SIRT1 overexpression also relieved the damage caused by CK40 to hepatocytes.

The effects of BG, EG, SG, EG + NK40, SG + NK40, EG + CK40, and SG + CK40 on the levels of proteins associated with the SIRT1/SREBP−1c/PGC−1α pathway in normal hepatocytes and in those treated with NEFAs at 2.4 mM were detected using Western blotting. Typical protein blot images are shown in Figure 6b,c. The results of gray-scale scanning for SIRT1/SREBP−1c/PGC−1α pathway protein levels in normal and 2.4 mM NEFA-administered hepatocytes overexpressing SIRT1 are shown in Figure 6d,e. The results showed that NK−derived EVs and overexpression of SIRT1 could both regulate the expression of SIRT1-SREBP−1c/PGC−1α pathway protein levels in normal calf hepatocytes, significantly inhibited the phosphorylation of AMPKα, significantly reduced the level of SREBP−1c protein, and increased the content of SIRT1 and PGC−1α protein. The overexpression of SIRT1, NK−derived EVs, and CK−derived EVs could regulate the expression of SIRT1/SREBP−1c/PGC−1α pathway proteins in 2.4 mM NEFA−treated calf hepatocytes. The overexpression of SIRT1 and NK−derived EVs might effectively alleviate liver lipid deposition and reduce lipid accumulation, while CK−derived exosome might aggravate liver lipid deposition and lipid accumulation.

### 2.8. Effects of SIRT1 Overexpression and Calf Hepatocyte EVs on SREBP−1c and PGC−1α Immunofluorescence and the Activities of Enzymes Related to the SIRT1/SREBP−1c/PGC−1α Pathway in Normal Hepatocytes and Those Treated with NEFAs at 2.4 mM

Figure 7a,b shows the immunofluorescence of PGC−1α and SREBP−1c in normal hepatocytes and those treated with NEFAs at 2.4 mM that were transfected with the adenovirus overexpressing SIRT1 and treated with EVs. In this study, the effects of BG, EG, SG, EG + NK40, SG + NK40, EG + CK40, and SG + CK40 treatments on SIRT1/SREBP−1c/PGC−1α pathway enzymes in normal hepatocytes and those treated with NEFAs at 2.4 mM were examined using ELISA. The results are shown in Figure 7c–f. For normal hepatocytes, overexpression of SIRT1 affected the activities of enzymes related to SIRT1/SREBP-1c/PGC−1α signaling and alleviated the effect of CK40 on normal hepatocytes. Overexpression of SIRT1 and NK40 both affected the activity of enzymes related to the SIRT1/SREBP-1c/PGC−1α pathway in 2.4 mM NEFA-treated calf hepatocytes. Overexpression of SIRT1 tended to reverse the changes in enzyme activity caused by CK40.

## 3. Discussion

For dairy cow, the most challenging stage is the transition period, characterized by alteration of the homeostatic state, increased risk of NEB, and metabolic disorders. During the transition period, an excess of 60% of dairy cow develops ketosis or fatty liver, comprising a direct response to an NEB, which might reduce liver efficiency, animal health, and milk yield [18]. Excessive NEB leads to fat mobilization, followed by an increase in the concentration of NEFAs and ketone bodies in the blood, which leads to disordered lipid metabolism of adipose tissue in calf with ketosis or fatty liver [19]. NEFAs play an important role in hepatic free fatty acid (FFA) metabolism and the concentration of NEFAs was significantly increased by increasing FFA activation and β−oxidation [20]. It has also been shown previously that NEFAs can cause a redox imbalance in cultured hepatocytes and disrupt hepatic lipid metabolism [21].

In previous studies, circulating body fluids were collected from cows and EVs were obtained according to different requirements. Limited by the individual differences among animals and the instability of model construction, scientific research into such EVs has not be carried out systematically. In this study, hepatocytes cultured in vitro were used to establish non-ketosis and clinical ketosis calf hepatocyte models. Subsequently, NK and CK EVs were obtained from the cell culture medium via ultra-high speed centrifugal. In this way, the EVs required for the test can be obtained in a short time. In this study, 2.4 mM NEFA treatment was used to simulate clinical ketosis of calf hepatocytes for exosome enrichment. There are two classes of characteristic exosome proteins. One is the transmembrane protein, which identifies the specific lipid–bilayer structure of the exosome; and the other cytosolic proteins, which indicate that the exosome can bind the membrane or transmembrane protein in the protoplasm [22]. Herein, we enriched the exosomes using differential ultra-high-speed centrifugation, and identified them via exosome surface marker proteins, such as CD63, CD9, and HSP70 [23]. In addition, the supernatant of the derived calf liver cells also contained exosome characteristic proteins, indicating that the cells release EVs into the extracellular environment; however, the concentration is insufficient when compared with the amount of enriched EVs. Studies have found that EVs are 20–200 nm saucer-shaped vesicles that are encapsulated in lipid bilayers without functional nuclei and cannot replicate freely [24]. In this study, TEM observed that the enriched EVs were typical saucer-shaped vesicles, which were dense, and had a diameter of 50−150 nm, in line with the characteristic appearance of EVs. The enriched CK−derived EVs had a relatively higher density than that of the NK−derived EVs. This is consistent with the observations in previous studies that more EVs are produced when tissues or cells are in a non-physiological state [25,26,27]. Intercellular communication-associated endosomes can be taken up by cells and then transmit information contained within the vesicles [28]. In this study, when the EVs were co-cultured with calf hepatocytes after labeling with PKH67, a large amount of granular PKH67 green fluorescence was observed inside the cells, indicating that the EVs could be ingested by the cells. In addition, we found that EVs enriched in the supernatant of hepatocytes could be taken up by other hepatocytes, suggesting that they might contain hepatocyte-linked signal transduction factors.

During the transition from late pregnancy to early lactation, large amounts of adipose tissue are mobilized in calf, resulting in elevated NEFA levels in the plasma, which can support the increased energy demand of the calf. Moreover, circulating NEFAs can be oxidized in hepatocytes or exported as a component of very-low-density lipoprotein [29]. The association of NEFAs with the regulation of AMPKα, SIRT1, PPARγ, PGC−1α was demonstrated by existing research results. AMPKα is a fuel-sensing protein that is activated in response to reduced cellular energy [30,31]. Hepatocyte-specific deletion of SIRT1 perturbed the normal metabolic function of PPARα engagement and reduced carboxylate β-oxidation, whereas overexpression of SIRT1 affected the interaction with PPARα and further influenced PGC−1 α activation. Indeed, it was reported that SIRT1 and PGC−1 α form stable complexes and that SIRT1 could regulate the activity and acetylation status of PGC−1α [32]. Trial evidence suggests that lipotoxicity, inflammation, ROS generation, endothelial dysfunction, and renal injury treated with a high-fat diet, are all associated with AMPK α activation [33]. PPARα activation of PGC−1 α/β to alleviate hepatic lipid metabolism disorders also inhibited the SREBP−1c pathway by reducing liver X recepto/retinoid X receptor formation, which is important in FFA metabolism regulation [34]. Depending on the state of exosome−derived cells or recipient cells, and the amount applied EVs, exogenous EVs can have beneficial or harmful effects on recipient cells or tissues [35]. However, the damaged cells in the disease state release EVs that have harmful effects on normal recipient cells, such as EVs from liver injury increasing recipient cell mortality [36]. In this study, NK−derived EVs increased SIRT1, AMPKα, PGC−1α, and ACO expression, increase the lipid conversion efficiency, and decrease the expression of SREBP−1c and FAS, thereby regulating hepatic lipid metabolism disorders and reducing lipid synthesis. Under the same test conditions, the CK−derived EVs had the opposite effect. Both exerted their most obvious effects at 40 μg/mL and when the concentration was higher, some indicators were not affected by EVs. This study showed that NK−derived EVs alleviated disorders of lipid metabolism in NEFA−treated hepatocytes by regulating SIRT1/SREBP−1c/PGC−1α−related genes, enzyme activities, and protein levels, even in normal hepatocytes. CK−derived EVs had a great effect on normal cultured calf hepatocytes and even regulated hepatocytes with disturbed hepatic lipid metabolism. During the transition period, it might be beneficial to improve the release of EVs by hepatocytes of normal calf or treat cows with NK−derived EVs to reduce or block the release of EVs from hepatocytes with hepatic lipid metabolism disorder via regulation of the SIRT1/SREBP−1c/PGC−1α pathway to ameliorate disordered hepatic lipid metabolism and provide energy for the body to reduce the NEB. While our findings provide valuable insights, certain methodological limitations must be duly acknowledged. The present study utilized primary hepatocytes derived from neonatal calves for all experiments. While calf hepatocytes offer practical advantages for primary cell isolation, we acknowledge their inherent limitations in fully replicating the metabolic state of periparturient calf. The use of neonatal cells was necessitated by ethical constraints on obtaining adult liver tissues from commercial dairy herds, though this may influence certain metabolic pathway responses compared to mature hepatocytes.

Activated expression of SIRT1, NAD+−dependent protein deacetylase, and AMPKα enhanced PGC−1α, followed by improved metabolism, mitochondrial activation, vascular regeneration, and increased cell survival [37,38]. SIRT1 is a member of the mammalian sirtuin family, whose effects include the inhibition of the nuclear NF-κB-dependent inflammatory response and influences gluconeogenesis, FFA oxidation, and mitochondrial biogenesis through PGC−1α [39]. PGC−1α, as well as SREBP, plays a crucial role in regulating lipid oxidation and lipid generation gene expression, thereby controlling the progression of hepatic steatosis, as well as later pathological development and different kinds of liver damage [40]. Thus, regulating SIRT1 can affect PGC−1α and SREBP−1c to regulate the SIRT1/SREBP−1c/PGC−1α pathway. In this study, SIRT1, AMPKα, and PGC−1α were upregulated by the overexpression of SIRT1 and NK−derived EVs. When the SIRT1 overexpressing adenovirus was transfected into 2.4 mM NEFA−treated calf hepatocytes, restoration of hepatocyte lipid metabolism was observed, which suggests a reduced risk of liver injury. However, the expression levels of SIRT1, AMPKα, and PGC−1α decreased under CK−derived EV treatment and the expression of SERBP−1c increased, contrary to the effect SIRT1 overexpression. In addition, there was no significant difference with the blank control group in liver cells overexpressing SIRT1, suggesting an antagonistic effect between the two. Thus, we concluded that both hepatocyte−derived EVs and SIRT1 overexpression could regulate hepatic lipid metabolism via SIRT-1/SREBP−1c/PGC−1α signaling. We further speculated that during the transition period, improving the release of EVs by normal liver cells might reduce or block the release of EVs produced during liver lipid metabolism disorders. In addition, regulation of the SIRT−1/SREBP−1c/PGC−1α pathway could reduce liver lipid metabolism disorders and provide energy to reduce the NEB. While our findings provide valuable insights, certain methodological limitations must be duly acknowledged. The present study utilized primary hepatocytes derived from neonatal calves for all experiments. While calf hepatocytes offer practical advantages for primary cell isolation, we acknowledge their inherent limitations in fully replicating the metabolic state of periparturient dairy cows. The main difference between liver cells of lactating cows and calves is that the former is adapted to high energy demand and lipid processing to support lactation, and compared with calf liver cells, the expression of genes for lipoprotein synthesis and fatty acid oxidation is increased. The use of calf hepatocyteneonatal cells was necessitated by ethical constraints on obtaining adult liver tissues from commercial dairy herds, though this may influence certain metabolic pathway responses compared to mature hepatocytes

In conclusion, the supernatant of hepatocytes cultured in vitro was enriched with EVs, which can be taken up by other hepatocytes as signaling factors in the liver. NK−derived and CK−derived EVs affected hepatocyte lipid metabolism in calf via SIRT1/SREBP−1c/PGC−1α signaling. The effect of SIRT1 overexpression on SIRT1/SREBP−1c/PGC−1α signaling in calf hepatocytes was consistent with the trend of NK−derived EVs and opposite to that of CK−derived EVs. In vitro assays confirmed that the effect of NK−derived EVs on the SIRT−1/SREBP−1c/PGC−1α pathway in calf hepatocytes in a high NEFA environment was consistent with the trend of SIRT1-overexpression and opposite to that of CK−derived EVs. The schematic diagram is shown in Figure 8. It is noteworthy that although this study is based on an in vitro model, the regulation of the SIRT1/SREBP−1c pathway by NK/CK−EVs shows remarkable consistency with the lipid metabolic abnormalities observed in clinical ketotic cows. Future research could validate the potential of these EVs as early diagnostic biomarkers for ketosis by detecting specific factors such as miRNA-27a in serum EVs from calf. Furthermore, investigating the impact of EV intervention on milk production performance would bridge mechanistic research with practical applications in dairy production.

## 4. Materials and Methods

### 4.1. Cells Culture

The Holstein calves needed for in vitro culture of calf hepatocytes were provided by a dairy farm in Hefei City, Anhui Province, China. The experimental procedures satisfied the requirements of the Scientific Ethics Committee of Anhui Agricultural University (Approval number 20210417). A two-step collagenase perfusion technique was used to harvest hepatocytes. It was anesthetized using thiamylal sodium and the caudate lobe of liver was removed. As previously described, hepatocytes were harvested using a two-step collagenase perfusion method [41]. Finally, the cells were cultured in a 5% CO_2_ thermostatic sterile cell incubator at 37 °C. The growth medium was changed every 24 h and the growth state of the cells was observed. The normal cultured hepatocytes of calf comprised the non−ketosis model group (NK). Isolated calf hepatocytes grown for 48 h to reach the optimal state, followed by culture with 3.8% Fetal Bovine Serum (FBS) (Sigma-Aldrich, St. Louis, MO, USA)for 12 h, and then treated for 9 h with NEFAs at 2.4 mM comprised the clinical ketosis (CK)model group (2.4 mM NEFA was selected based on the blood NEFA content of cows with ketosis and experimental results) [42]. The NEFA treatment concentration of 2.4 mM was selected based on the following rationale: reported that this concentration corresponds to the average blood NEFA levels observed in clinically ketotic calf (2.1–2.6 mM range) [42], and our preliminary dose-response experiments confirmed this concentration effectively induced ketogenic metabolic alterations without causing excessive cytotoxicity.

### 4.2. EV Isolation and Characterization

In this study, EVs were obtained from cell culture supernatants of the following two cell models: the NK group (0 mM NEFA−treated calf hepatocytes) and the CK group (2.4 mM NEFA-treated calf hepatocytes). We collected the supernatants of the cell cultures of both groups and extracted the EVs using differential ultra-high-speed centrifugation. The supernatants of cultured NK and CK cells were centrifuged at 4 °C for 30 min to remove dead cells. The supernatants were then centrifuged at 4 °C for 2 h at 20,000× *g* to remove cell debris, followed by centrifugation at 4 °C for 2 h at 200,000× *g* in a supporting ultracentrifuge tube. The EV suspensions were obtained by discarding the supernatant and resuspending the precipitates in 50–200 μL PBS. The above test flow is shown in Figure 1a.

### 4.3. Nanoparticle Tracking Analysis (NTA)

The Flow NanoAnalyzer U30E (NanoFCM, Xiamen, China) was used to measure the number and size of EVs using NTA. The instrument was calibrated using 100 nm polystyrene beads before use. The concentration (particle/mL) and size distribution of nanoparticles were measured employing the Flow NanoAnalyzer U30E. For each sample, the NTA software (ZetaView 8.04.02 SP2) batch process was utilized to integrate the three technical indicators. The as analyzed average vesicle size and concentration were corrected by the dilution factor as required. Each sample was analyzed at least three times.

### 4.4. Transmission Electron Microscopy (TEM)

To clearly observe the morphology of the EVs and protect the instrument, only EV samples with low levels of impurities and with a suitable concentration were used. When the EVs were extracted by ultra-high−speed centrifugation, we used PBS to wash them again, and then resuspended them in PBS to ensure that the EVs could be adsorbed onto the copper mesh without breaking due to their excessive concentration. The specific steps were as follows: the EVs were suspended in 200 μL of PBS. The Cu mesh was placed face down on the EV sample solution, and allowed to absorb for 5 min. Excess liquid was removed and the mesh was allowed to dry. Then, the copper mesh was incubated with 2% phosphotungstic acid dyeing solution for 50 s, after which the dyeing solution was removed and the mesh allowed to dry. The samples were then viewed under a transmission electron microscope at 80 kV.

### 4.5. Fusion of EVs and Calf Hepatocytes

The enriched EV precipitate was resuspended in 45 μL diluent C (A sterile water-soluble solution), added with 5 μL of PKH67 (250×) (a green fluorescent label for cell membranes) (Beyotime, Shanghai, China) and 1.2 mL serum free medium incubated in the dark at room temperature for 4 min, added with growth medium to stop staining, and centrifuged at high speed (200,000× *g*) for 1 h to remove residual dyes. The obtained EVs were resuspended and stored in the dark at 4 °C for subsequent uptake experiments.

Cells were seeded in 24-well plates and incubated with PKH67-labeled EVs at 37 °C for 12 h. The cells in each well were fixed using 4% paraformaldehyde in the dark for 20 min and then incubated with Hoechst 33342 (a fluorescent DNA stain) (Sigma-Aldrich) in the dark for 5 min. Finally, the cell on slides were removed from the 24-well plates and sealed with an anti-fluorescence quenching agent. The fusion of EVs and calf hepatocytes was observed using laser confocal microscopy.

### 4.6. Cell Viability Assay

When they reached the logarithmic growth phase, calf hepatocytes were added to the wells of a 96-well plate at 1 × 10^5^ cells/mL, cultured under normal or high NEFA conditions (cultured with 3.8% FBS for 12 h and then NEFAs were added to a final concentration 2.4 mM, followed by culture for 9 h). Separately, NK and CK EVs were added to the final concentrations 0, 20, 40, 60 and 80 μg/mL in the medium, followed by culture for 12 and 24 h. Cell counting kit 8 (CCK-8) (Beyotime, Shanghai, China) assays were employed to ascertain the optimal concentration of EVs. The assessment was repeated three times.

### 4.7. Construction of the SIRT1-Overexpressing Adenovirus

The AdMAX adenovirus packaging system was used in this experiment (SignaGen Laboratories Frederick, Maryland Co., USA). The system comprises the adenovirus backbone vector pBHGloxE1,3cre and the pHBAd shuttle plasmid. To construct the SIRT1 overexpressing adenovirus, we first constructed the SIRT1 shuttle plasmid and the adenovirus skeleton vector. Then, the recombinant backbone plasmid pHBAd-BHG and adenovirus vector plasmid h-SIRT1 were mixed and transfected into HEK293 cells using LipofiterTM transfection reagent (Hanbio Biotechnology, Shanghai, China) for recombination. When the majority of the HEK293 cells displayed cytopathic effects, the cell culture solution was collected for the first time. The solution was frozen and thawed repeatedly (three times) between a 37 °C water bath and liquid nitrogen, followed by centrifugation for 5 min at 1000× *g* to collect the supernatant, which contained the first generation of virus species. Some of the first-generation virus species were reinfected into HEK293 cells and the rest were reserved at −80 °C. The second-generation viruses were collected after the observation of cytopathic effects. Subsequently, the above method was used to obtain the third-generation virus. Thereafter, the SIRT1 overexpressing adenovirus and the control virus were purified via double CsCl density gradient 100,000× g ultracentrifugation and diluted twice using Tris-EDTA buffer solution. The purified viruses were dialyzed three times against 200× dialysis buffer, with the liquid being changed every hour. Finally, the adenovirus titer was determined by the 50% tissue culture infection method to make the multiplicity of infection (MOI) = 100 and stored in a −80 °C refrigerator [43].

### 4.8. Quantitative Real Time Reverse Transcription Polymerase Chain Reaction (qRT-PCR)

The TRI reagent (T9424; Sigma-Aldrich) was employed to extract total RNA from cells. cDNA was reverse transcribed from the total RNA using a superscript III reverse transcription kit (Invitrogen, Waltham, MA, USA). Bioengineering Co., Ltd. (Shanghai, China) designed and synthesized the primer sequences for the target genes and the *ACTB* (encoding β-actin) reference gene, according to the sequences stored in GenBank (Table 1). The cDNA was used as the template for amplification using an ABI 7500TM Fast qPCR instrument (Thermo Fisher Scientific, Waltham, MA, USA), employing the parameter settings: 95 °C pre-denaturation for 2 min; 95 °C for 20 s, 60 °C for 20 s, 72 °C for 30 s, 40 cycles. The results were expressed using the 2^−∆∆Ct^ method and each sample was repeated three times.

### 4.9. Enzyme Linked Immunosorbent Assay (ELISA)

Calf hepatocytes in the logarithmic growth phase were inoculated into the wells of six-well plates at 2 × 10^5^ cells/mL and grown for 24 h. Treatments were then caried out according to the experimental grouping. Three cell supernatants were collected from each group and centrifuged at 2000–3000 r/min for 20 min. The enzymatic activities of SIRT1 (No. ml037854), phosphorylated (p)AMPKα (No. ml063281), SREBP−1c (No. ml037082), and PGC−1α (No. ml058244) in the cell supernatants were detected using ELISA kits (Shanghai Enzyme-linked Biotechnology Co., Ltd., Shanghai, China) with reference to the kit instructions.

### 4.10. Western Blotting Assays

Western blotting was carried out following a previously published protocol [31]. Antibodies recognizing cytokeratin 18 (CK18) (Proteintech, Rosemont, IL, USA), CD9 (Affinity Biosciences, Cincinnati, OH, USA), CD63 (Affinity Bioscience), heat shock protein 70 (HSP70) (Affinity Bioscience), SREBP−1c, PGC1α, SIRT1 (Novus Biologicals, Centennial, CO, USA), and AMPKα, pAMPKα (Cell Signaling Technology, Danvers, MA, USA)were used for detection. A bicinchoninic acid protein concentration kit (Beyotime Biotechnology, Shanghai, China) was employed to determine the EV protein concentrations.

### 4.11. Immunofluorescence

Hepatocytes in the logarithmic growth phase were inoculated into the wells of a 24-well plate at 1 × 10^5^ cells/mL and cultured for 24 h. The cells were grouped according to the test conditions. The cells in each well were fixed using 4% paraformaldehyde in the dark for 20 min, made permeable using 0.5% Triton × 100 at room temperature for 20 min, and subjected to 0.5% BSA blocking at room temperature for 30 min. The primary antibodies (SREBP−1c and PGC−1α), diluted in 0.5% BSA, were then added and incubated overnight at 4 °C. Next day, the samples were incubated with fluorescein isothiocyanate (FITC)fluorescently labeled secondary antibodies for 50 min, and then stained using 4′,6-diamidino-2-phenylindole (DAPI) (Beyotime Biotechnology, Shanghai, China) for 6 min. The cells on slides were taken out of the 24-well plates and sealed with an anti-fluorescence quenching agent. Laser scanning confocal microscopy was then used to observe the cells on the slides.

### 4.12. Statistical Considerations

All experimental data are presented as the mean ± SD. We carried out ANOVA using SPSS Statistics 20.0 statistical software (IBM Corp., Armonk, NY, USA), followed by Tukey’s post hoc test for multiple comparisons. We utilized GraphPad Prism 7.0 software (GraphPad, La Jolla, CA, USA) for generating statistical annotations on histograms and line graphs, as well as for analyzing and smoothing the size distribution of extracellular vesicles (EVs). The clipping and splicing of electron microscope photographs were performed using Adobe Photoshop CC 2017 software (San Jose, CA, USA). The *p* value between the two groups was obtained by independent *T*-test. * *p* < 0.05 was considered to indicate statistical significance and ** *p* < 0.01 was considered to indicate extreme significance.

## Figures and Tables

**Figure 1 ijms-26-09392-f001:**
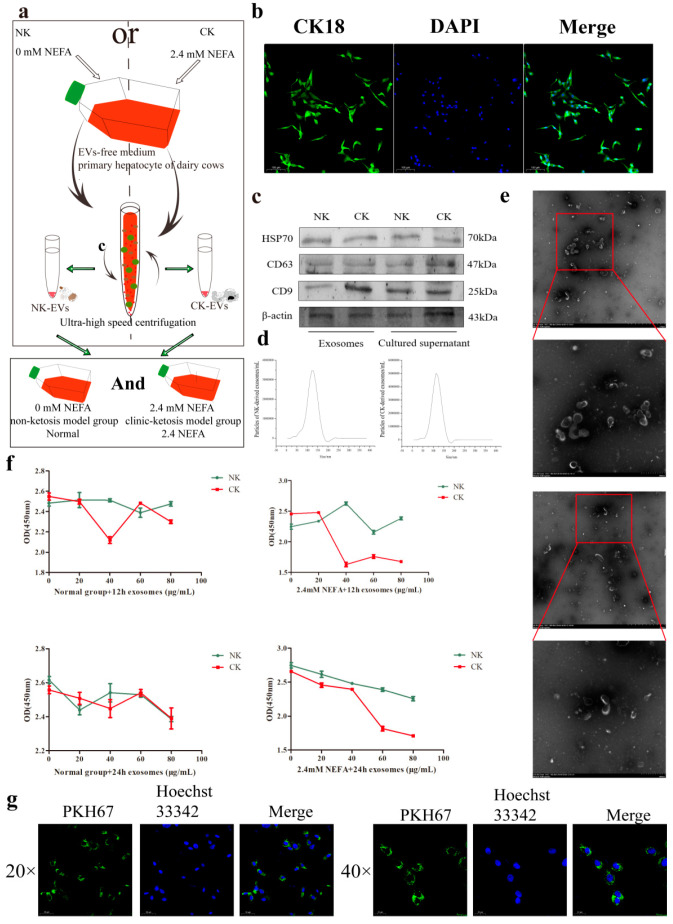
Characterization and fusion of exosomes from cultured calf hepatocytes. (**a**) Test flow and pattern diagram. (**b**) Identification of calf liver cell surface proteins using immunofluorescence. (**c**) Identification of EV surface marker proteins using Western blotting. (**d**) The size distribution of EVs was detected using NTA. (**e**) The morphology of EVs as observed using a transmission electron microscope. (**f**) Effect of a gradient concentration of EVs on hepatocytes activity. NK, Non-ketosis. CK, Clinical ketosis. (**g**) Immunofluorescence analysis of hepatocytes that internalized PKH67-labeled EVs. Typical images of Western blotting are shown. Results are presented as the mean ± SD (*n* = 3).

**Figure 2 ijms-26-09392-f002:**
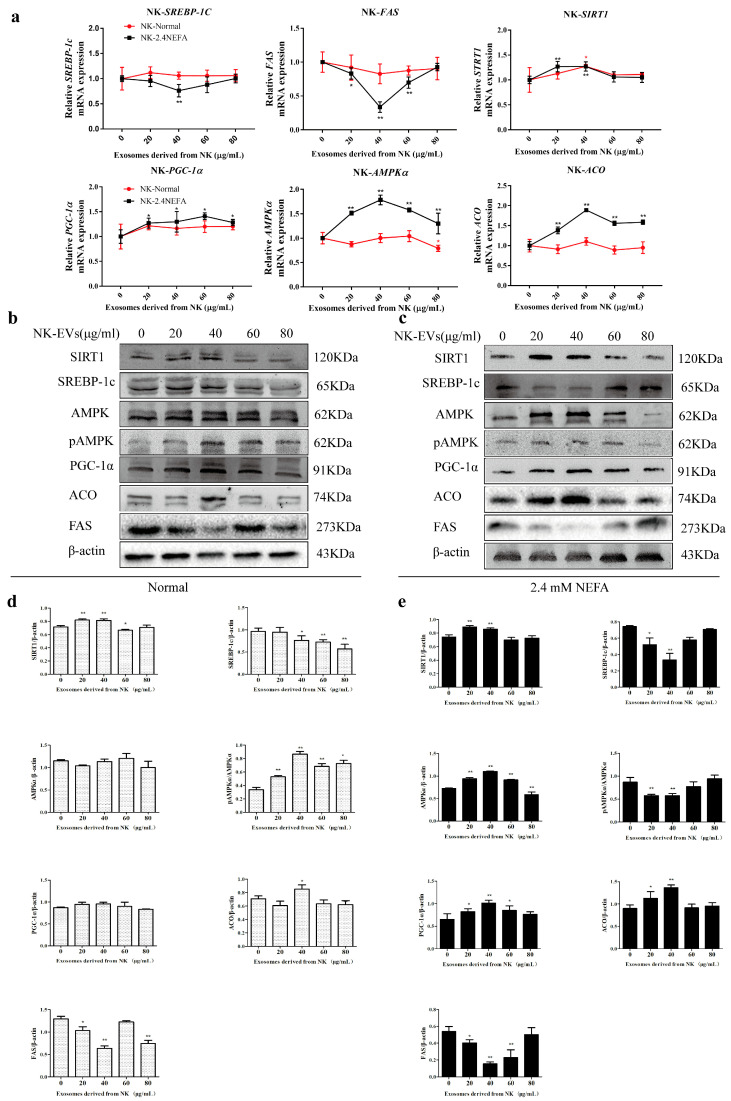
NK−derived EVs modulate the relative mRNA and protein expression levels of genes related to the SIRT1/SREBP−1c/PGC−1α pathway in calf hepatocytes. (**a**) Effects of NK−derived EVs on the relative mRNA expression levels of SIRT1-SREBP−1c/PGC−1α pathway-related genes in untreated (normal) calf hepatocytes and those treated with NEFAs at 2.4 mM. (**b**,**c**) Effects of NK−derived EVs on the relative levels of SIRT1/SREBP−1c/PGC−1α pathway-related proteins in normal calf hepatocytes and those treated with NEFAs at 2.4 mM. (**d**,**e**) The relative protein levels were quantified in normal calf hepatocytes and those treated with NEFAs at 2.4 mM NEFA using ImageJ 2.0.0−rc−30. The level of β-actin was used to normalize the relative protein expression. NK, Non-ketosis. NK-Normal, normal calf hepatocytes treated with NK-derived EVs. NK−2.4 mM NEFA, treated with NK−derived EVs and NEFAs at 2.4 mM. Results are presented as the mean ± SD (*n* = 3). * *p* < 0.05 and ** *p* < 0.01 vs. 0 μg/mL NK−derived EVs. Typical images of Western blotting are shown.

**Figure 3 ijms-26-09392-f003:**
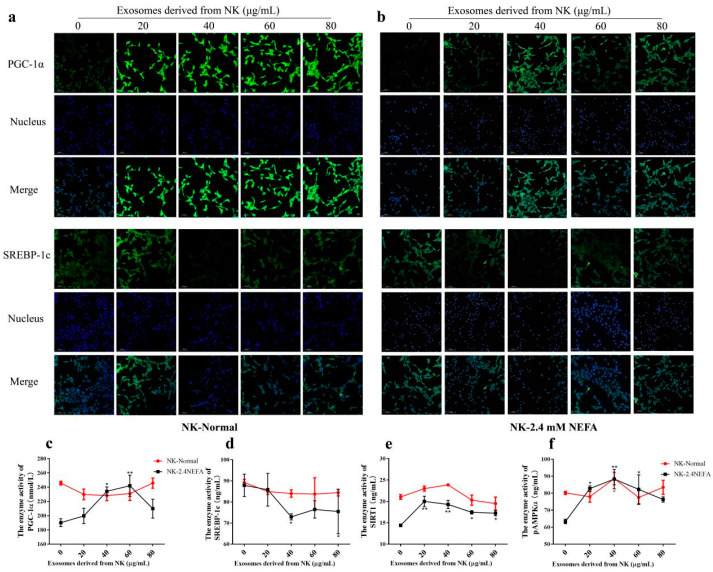
Effects of NK−derived EVs on SIRT1/SREBP−1c/PGC−1α pathway−related proteins and enzyme activities. (**a**,**b**) Immunofluorescence assay showing the effect of NK−derived EVs on SREBP−1c and PGC−1α proteins in normal and 2.4 mM NEFA−treated hepatocytes. (**c**–**f**) Effects of NK−derived EVs on SIRT1/SREBP−1c/PGC−1α pathway−related enzyme activities in normal and 2.4 mM NEFA−treated hepatocytes. NK, Non−ketosis. NK−Normal, normal hepatocytes treated with NK−derived EVs. NK−2.4 mM NEFA, hepatocytes treated with NK−derived EVs and NEFAs at 2.4 mM. Results are presented as the mean ± SD (*n* = 3). * *p* < 0.05 and ** *p* < 0.01 vs. 0 μg/mL NK−derived EVs.

**Figure 4 ijms-26-09392-f004:**
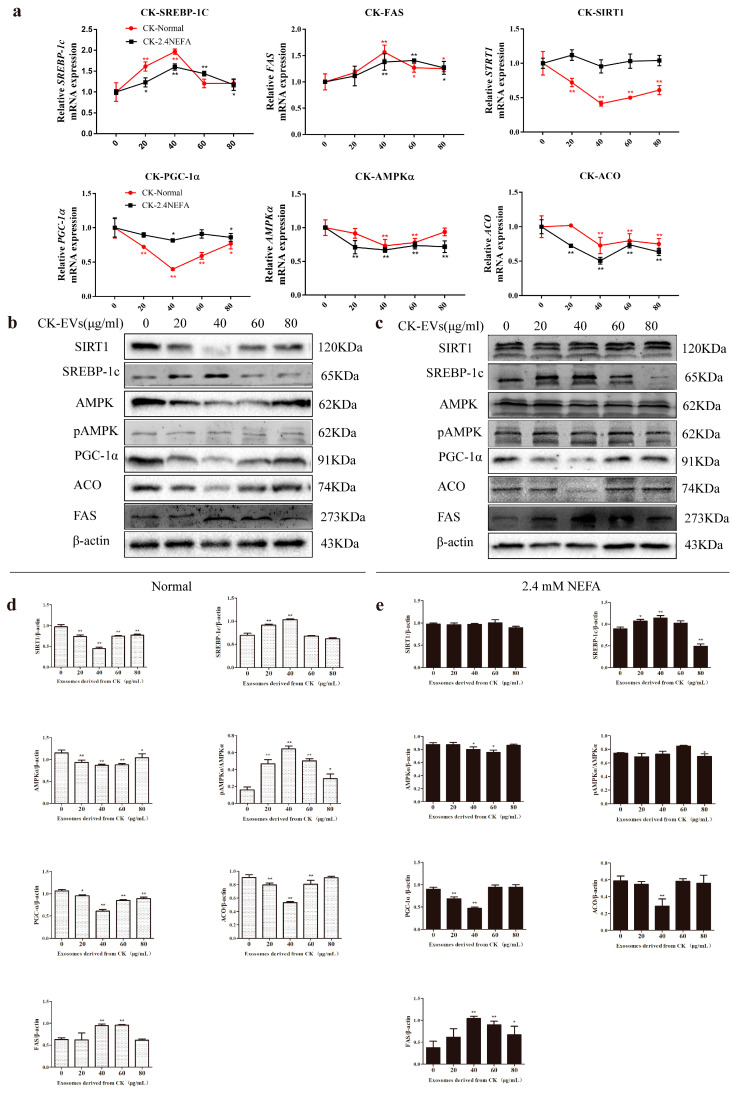
CK−derived EVs modulate the mRNA and protein expression levels of SIRT1/SREB−1c/PGC−1α pathway-related genes in hepatocytes. (**a**) Effects of CK−derived EVs on the relative mRNA expression levels of SIRT1-SREBP−1c/PGC−1α pathway-related genes in normal and 2.4 mM NEFA−treated hepatocytes. (**b**,**c**) Effects of CK−derived EVs on the relative abundance of SIRT1/SREBP−1c/PGC−1α pathway-related proteins in normal hepatocytes and those treated with NEFAs at 2.4 mM. (**d**,**e**) The relative protein levels were quantified in the normal and 2.4 mM NEFA groups using ImageJ 2.0.0−rc−30. The levels of β-actin were used to normalize the relative protein levels. CK, Clinical ketosis. CK−Normal, normal hepatocytes treated with CK−derived EVs. CK-2.4 mM NEFA, hepatocytes treated with CK−derived EVs and NEFAs at 2.4 mM. Results are presented as the mean ± SD (*n* = 3). * *p* < 0.05 and ** *p* < 0.01 vs. 0 μg/mL CK−derived EVs. Typical images of Western blotting are shown.

**Figure 5 ijms-26-09392-f005:**
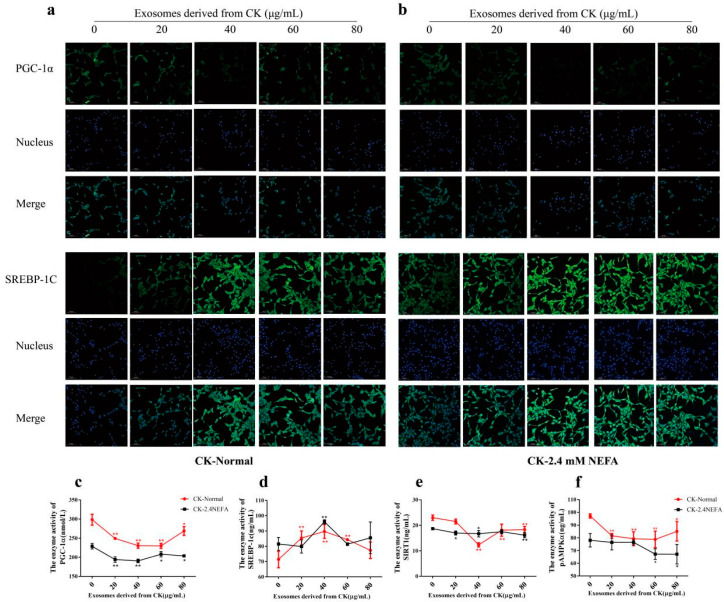
Effects of CK−derived EVs on SIRT1/SREBP−1c/PGC−1α pathway-related proteins and enzyme activities. (**a**,**b**) Immunofluorescence assay showing the effect of CK−derived EVs on SREBP−1c and PGC−1α proteins in normal and 2.4 mM NEFA-treated hepatocytes. (**c**−**f**) Effects of CK−derived EVs on SIRT1/SREBP−1c/PGC−1α pathway-related enzyme activities in normal and 2.4 mM NEFA−treated hepatocytes. CK, Clinical ketosis. CK-Normal, normal hepatocytes treated with CK−derived EVs. CK-2.4 mM NEFA, hepatocytes treated with CK−derived EVs and 2.4 mM NEFA. Results are presented as the mean ± SD (*n* = 3). * *p* < 0.05 and ** *p* < 0.01 vs. 0 μg/mL CK−derived EVs.

**Figure 6 ijms-26-09392-f006:**
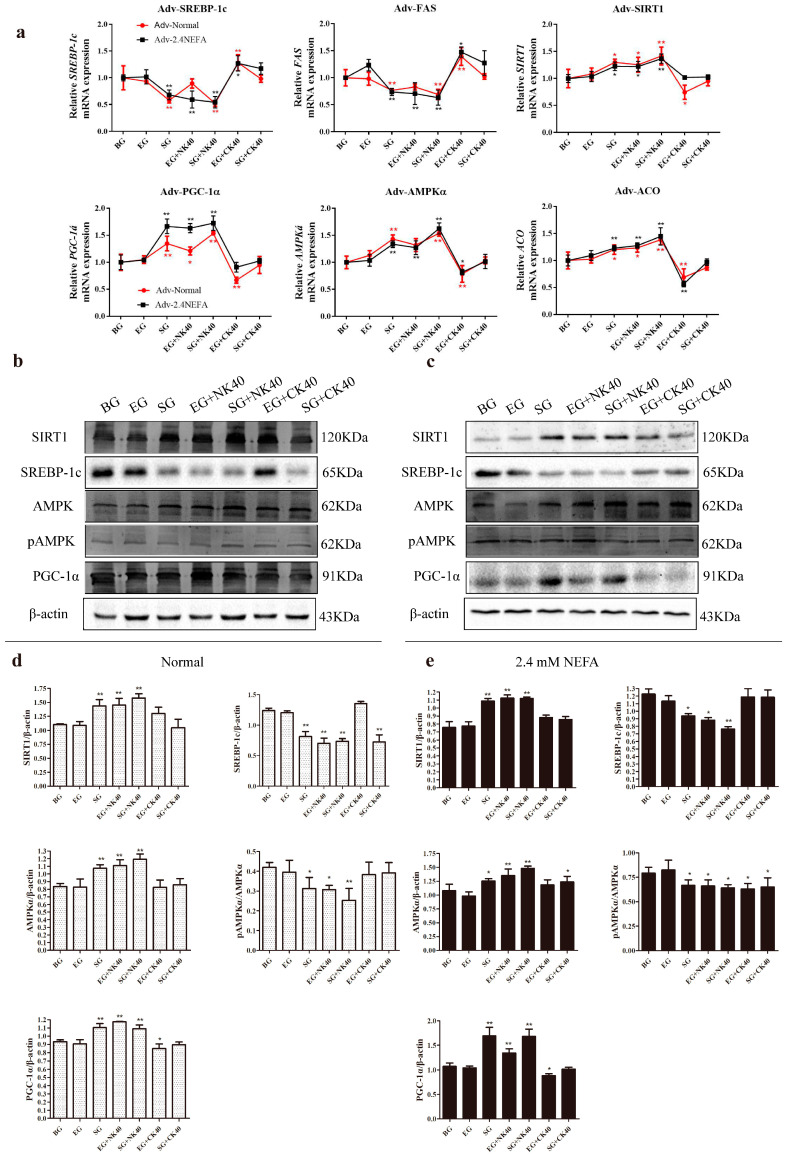
SIRT1 overexpression and EVs modulate the relative mRNA and protein expression levels of SIRT1/SREBP−1c/PGC−1α pathway−related genes in hepatocytes. (**a**) Effects of SIRT1 overexpression and EVs on mRNA relative abundance of SIRT1−SREBP−1c/PGC−1α pathway−related genes in normal and 2.4 mM−NEFA treated hepatocytes. (**b**,**c**) Effects of SIRT1 overexpression and EVs on the relative abundance of SIRT1/SREBP−1c/PGC−1α pathway-related proteins in untreated (normal) hepatocytes and those treated with NEFAs at 2.4 mM. (**d**,**e**) The relative protein levels were quantified in the normal hepatocytes and the cells treated with NEFAs at 2.4 mM using ImageJ 2.0.0−rc−30. The level of β-actin was used to normalize the relative protein expression. NK, Non-ketosis. CK, Clinical ketosis. Results are presented as the mean ± SD (*n* = 3). * *p* < 0.05 and ** *p* < 0.01 vs. the blank group (BG). Empty carrier group (EG); SIRT1-overexpression group (SG); 40 μg/mL NK−derived EVs (NK40); 40 μg/mL CK−derived EVs (CK40). Typical images of Western blotting are shown.

**Figure 7 ijms-26-09392-f007:**
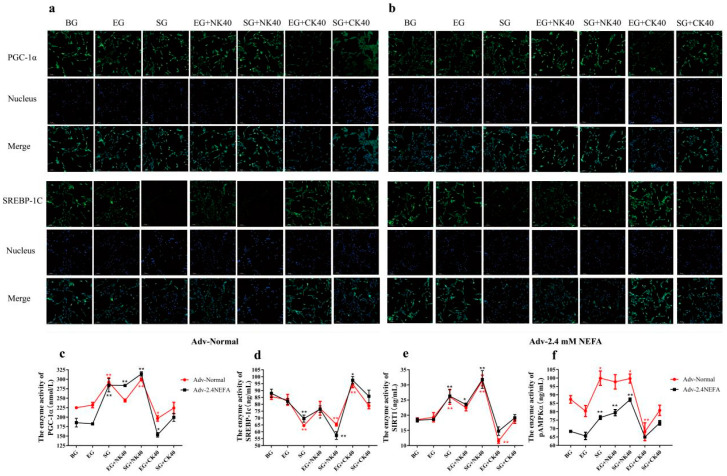
Effects of EVs and the SIRT1-overexpressing recombinant adenovirus on SIRT1/SREBP-1c/PGC−1α pathway-related proteins and enzyme activities. (**a**,**b**) Immunofluorescence assay showing the effect of EVs and SIRT1 overexpression on SREBP−1c and PGC−1α proteins in normal and 2.4 mM NEFA-treated hepatocytes. (**c**–**f**) Effects of EVs and SIRT1 overexpression on SIRT1/SREBP−1c/PGC−1α pathway related enzyme activities in normal and hepatocytes receiving NEFAs at 2.4 mM. NK, Non-ketosis. CK, Clinical ketosis. Results are presented as the mean ± SD (*n* = 3). * *p* < 0.05 and ** *p* < 0.01 vs. the blank group (BG). Empty carrier group (EG); SIRT1−overexpression group (SG); 40 μg/mL NK−derived EVs (NK40); CK40, 40 μg/mL CK−derived EVs (CK).

**Figure 8 ijms-26-09392-f008:**
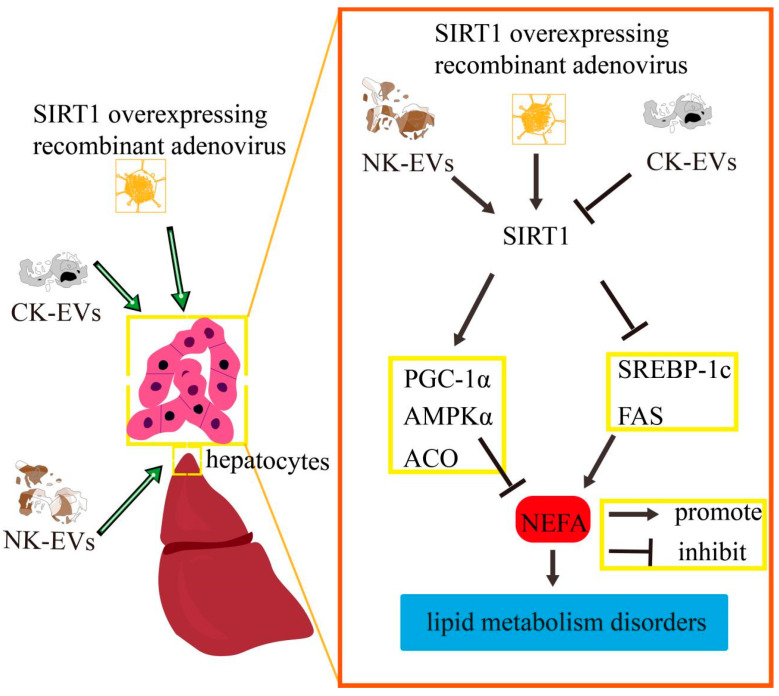
Mechanism by which EVs regulate hepatic lipid metabolism via SIRT1-SREBP−1c/PGC-1α. NK−derived EVs and SIRT1 overexpression suppressed the interference of high NEFA levels on hepatocyte lipid metabolism in calf. In contrast, CK−derived EVs exacerbated this effect through this pathway. NK, Non-ketosis. CK, clinical ketosis.

**Table 1 ijms-26-09392-t001:** Primer sequences for qPCR analysis.

Genes	Primer Sequences	Accession Number	Product Size/bp
*SIRT1*	F: GCTTACAGGGCCTATCCAG R: CATGCGAGGCTCTATCATCT	NM_001192980.2	186
*AMPKα*	F: ACCAAGGTGTAAGGAAAGCA R: ACGGGTTTACAACCTTCCAT	NM_001109802.2	126
*FAS*	F: TTCTTAGACAAGCCCCTCTC R: TAGGTAGTTCGGAGCATCTG	NM_174662.2	150
*ACO*	F: AGACCACTATTACAAGGCCG R: AATACGTGCATGTGTGGTTG	NM_001205495.1	127
*PGC1A*	F: GCCCCAGGTGGTGGA R: GTTACTTTCCAGAGGAGGCA	NM_177945.3	109
*SREBP1C*	F: GCTGACCGACATAGAAGACAT R: CCAGGAAGCCTTCAAGTGAG	NM_001113302.1	188
*ACTB*	F: GCCCTGAGGCTCTCTTCCA R: GCGGATGTCGACGTCACA	NM_173979.3	100

## Data Availability

Study data are available to the corresponding author by email.

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
