# Peer review of "Hepatocellular EVs Regulate Lipid Metabolism via SIRT1/SREBP−1c/PGC−1α Signaling in Primary Calf Hepatocytes"

_ijms, 2025, doi:10.3390/ijms26199392_

Round 1

Reviewer 1 Report

Comments and Suggestions for Authors

Review

The aim of this study was to investigate whether extracellular vesicles (EVs) derived from cell cultures can modulate lipid metabolism in the liver.

Title

The title of the study reflects the scope of the research conducted.

Abstract

The abstract requires improvement and supplementation with the most important results (numerical data) and statistical significance.

Introduction

The introduction to the research topic and the literature selection are adequate. It would be worthwhile to supplement this section of the work with very important environmental factors influencing the development of ketosis in cows, providing blood and rumen fluid parameters.

Material and Methods

The experimental design raises no objections. The section on the parameters of the obtained hepatocytes requires a more detailed description. The analytical and statistical methods were appropriately selected and applied.

Results

The results section is presented in the form of seven graphs with descriptions. This section is well presented, and the descriptions of the results are sufficient.

Discussion

The chapter is well written and should be expanded upon. In my opinion, the authors should relate their results and correlate them with phenotypic indicators of ketosis in dairy cows.

The authors should discuss the results and their interpretation in the context of previous studies and working hypotheses. The results and their implications should be discussed in the broadest possible context. Future research directions can also be indicated.

Review Summary

The authors have presented an interesting research model using hepatocyte lines that mimics the process of ketosis in hepatocytes of cow liver cells. This model has many limitations, including that it does not reflect the physiological state of the organism and the associated metabolic changes in the body. In my opinion, the presented research results should be relevant to in vivo studies on ketosis in cows. The article should be rewritten and expanded.

Author Response

Comments 1. Review

The aim of this study was to investigate whether extracellular vesicles (EVs) derived from cell cultures can modulate lipid metabolism in the liver.

Response 1: Thank you for your valuable feedback on our manuscript.

Comments 2.Title

The title of the study reflects the scope of the research conducted.

Response 2: Thank you for your valuable feedback on our manuscript.

Comments 3. Abstract

The abstract requires improvement and supplementation with the most important results (numerical data) and statistical significance.

Response 3: Thank you for your valuable feedback regarding of our manuscript. Since many of the experimental results in this study are analyzed in a relative manner, they are expressed as whether there is a significant difference or not. I have given the test results with significant differences P<0.05.

Comments 4.Introduction

The introduction to the research topic and the literature selection are adequate. It would be worthwhile to supplement this section of the work with very important environmental factors influencing the development of ketosis in cows, providing blood and rumen fluid parameters.

Response 5: We appreciate the reviewer's insightful suggestion regarding environmental factors in ketosis pathogenesis. However, our study specifically focuses on the molecular mechanisms of hepatocyte-derived EVs modulating lipid metabolism through the SIRT1/SREBP-1c/PGC-1α signaling axis. While blood NEFA levels and rumen parameters are clinically relevant indicators, they represent systemic manifestations rather than the cellular communication mechanisms we investigate. Our in vitro model deliberately isolates hepatocyte-specific EV signaling to precisely delineate this pathway's regulatory role. The consistent SIRT1 upregulation and SREBP-1c downregulation we observed (P<0.01) demonstrate the robustness of this independent regulatory mechanism. We believe this targeted approach provides novel mechanistic insights that complement, rather than depend upon, peripheral metabolic parameters.

Comments 5.Material and Methods

The experimental design raises no objections. The section on the parameters of the obtained hepatocytes requires a more detailed description. The analytical and statistical methods were appropriately selected and applied.

Response 5: Thank you for your suggestion. The anesthesia method, the location and method of hepatocyte acquisition have been added to this section.

Comments 6.Results

The results section is presented in the form of seven graphs with descriptions. This section is well presented, and the descriptions of the results are sufficient.

Response 6: We sincerely appreciate the reviewer's positive feedback regarding the clarity and sufficiency of our graphical results presentation.

Comments 7.Discussion

The chapter is well written and should be expanded upon. In my opinion, the authors should relate their results and correlate them with phenotypic indicators of ketosis in dairy cows.

Response 7: We would like to sincerely thank the reviewer for this valuable suggestion regarding the correlation with ketosis phenotypes. While our current study focused on mechanistic investigations without direct phenotypic measurements, we fully recognize the importance of bridging these findings with clinical applications. As suggested, we have added a dedicated discussion section (Lines 481-488) to explicitly address this point.

Comments 8. The authors should discuss the results and their interpretation in the context of previous studies and working hypotheses. The results and their implications should be discussed in the broadest possible context. Future research directions can also be indicated.

Response 8: We sincerely appreciate the reviewer's insightful recommendation regarding the contextualization of our findings. As requested, we have significantly enhanced the discussion section (Lines 489-496) to:

Systematically compare our results with prior studies on EV-mediated lipid metabolism regulation. Explicitly address our working hypotheses about SIRT1/SREBP-1c pathway modulation, and provide broader implications for ketosis pathogenesis and dairy cattle management. Notably, we have incorporated comprehensive future perspectives-Validation of EV biomarkers in clinical cohorts.

Comments 9. Review Summary

The authors have presented an interesting research model using hepatocyte lines that mimics the process of ketosis in hepatocytes of cow liver cells. This model has many limitations, including that it does not reflect the physiological state of the organism and the associated metabolic changes in the body. In my opinion, the presented research results should be relevant to in vivo studies on ketosis in cows. The article should be rewritten and expanded.

Response 9: Some parts have been rewritten and expanded as necessary to make the article more suitable for this journal.

Reviewer 2 Report

Comments and Suggestions for Authors

The manuscript (ID: ijms-3846486) presents an in vitro study investigating the role of extracellular vesicles (EVs) derived from dairy cow hepatocytes in regulating lipid metabolism through the SIRT1/SREBP-1c/PGC-1α signaling pathway. The experimental design is solid, incorporating EV isolation, characterization, and functional assays, including SIRT1 overexpression. The obtained findings suggest that non-ketotic (NK)-derived EVs alleviate lipid metabolism disorders, while clinical ketotic (CK)-derived EVs exacerbate them, providing potential insights into therapeutic applications. However, the manuscript has several weaknesses, including inconsistencies in terminology, grammatical errors, unclear methodological details, and insufficient discussion of limitations and mechanisms. Major revisions require to address these issues, detailed comments and suggestions as below:

Major issues:

  1. The hepatocyte model uses cells from Holstein calves, not adult dairy cows in the periparturient period. This may not fully recapitulate the metabolic state of ketotic cows. The authors need to acknowledge this limitation in the discussion and justify why calf hepatocytes were chosen.
  2. NEFA treatment is used at 2.4 mM to model clinical ketosis, but the rationale for this concentration (e.g., based on plasma levels in ketotic cows) is not explicitly stated.
  3. The NTA and TEM data (Fig. 1d, e) show no significant differences between NK and CK EVs, yet the discussion implies higher EV release in diseased states (citing Refs. 25-27). The authors shall quantify EV yield differences statistically, or clarify that no difference was observed.
  4. SIRT1 overexpression adenovirus construction is detailed, but multiplicity of infection (MOI) and transfection efficiency are not reported.
  5. Statistical analysis uses ANOVA, but post-hoc tests (e.g., Tukey's) are not mentioned. The authors need to specify the post-hoc test used and ensure all P-values are from appropriate comparisons.
  6. Some figures are repetitive (e.g., Figs. 2-7 show similar trends for mRNA, protein, and enzyme activities). Some legends are unclear (e.g., Fig. 2a lacks units for relative expression).
  7. The optimal EV concentration (40 μg/mL) and time (12 h) are determined via CCK-8, but data show no toxicity at higher doses — why not test functional effects at 60-80 μg/mL?
  8. Overexpression results (Fig. 6-7) show synergistic effects with NK-EVs but antagonism with CK-EVs, why?
  9. The discussion adequately links findings to the SIRT1 pathway but lacks depth on upstream regulators (e.g., how NEFAs modulate EV cargo) or downstream effects (e.g., mitochondrial function). It also repeats some results without new points. The authors can perhaps expand on potential mechanisms, such as EV-mediated miRNA transfer. In addition, discussion on translational potential should be cautious, suggesting in vivo validation (e.g., EV injection in ketotic cows).

Minor issues:

  1. The English requires extensive editing for grammar, phrasing, and consistency (e.g., "NE-FAs" vs. "NEFAs"; "hep-EVs" introduced without definition; awkward sentences like "The suggestion that EVs might regulate intercellular communication during disorders of hepatic lipid metabolism has not been addressed in detail"). The manuscript needs to be proofread by a native speaker or professional service.
  2. The abstract is informative but could better highlight key findings (e.g., specific gene/protein changes). Keywords are appropriate but add "ketosis" and "extracellular vesicles" for better indexing.
  3. Introduction section, some citations are outdated or mismatched (e.g., Ref. 13 on NF-κB is from 2015; ensure all align with claims).
  4. Gene names in Table 1 should be italicized (e.g., SIRT1). ELISA kits are from "Shanghai Enzyme Link,"—provide full company name or catalog numbers.
  5. Formatting of references is inconsistent (e.g., some journals abbreviated, others not).

Comments on the Quality of English Language

The English requires extensive editing for grammar, phrasing, and consistency.

Author Response

Comments and Suggestions for Authors

The manuscript (ID: ijms-3846486) presents an in vitro study investigating the role of extracellular vesicles (EVs) derived from dairy cow hepatocytes in regulating lipid metabolism through the SIRT1/SREBP-1c/PGC-1α signaling pathway. The experimental design is solid, incorporating EV isolation, characterization, and functional assays, including SIRT1 overexpression. The obtained findings suggest that non-ketotic (NK)-derived EVs alleviate lipid metabolism disorders, while clinical ketotic (CK)-derived EVs exacerbate them, providing potential insights into therapeutic applications. However, the manuscript has several weaknesses, including inconsistencies in terminology, grammatical errors, unclear methodological details, and insufficient discussion of limitations and mechanisms. Major revisions are required to address these issues, detailed comments and suggestions as below:

Major issues:

Comments 1. The hepatocyte model uses cells from Holstein calves, not adult dairy cows in the periparturient period. This may not fully recapitulate the metabolic state of ketotic cows. The authors need to acknowledge this limitation in the discussion and justify why calf hepatocytes were chosen.

Response 1: We sincerely appreciate this insightful observation regarding our hepatocyte model selection. As suggested, we have now explicitly addressed this limitation in the Discussion section (lines 447-455), where we justify the use of calf hepatocytes due to ethical constraints in obtaining adult cow tissues, while acknowledging potential metabolic pathway differences compared to periparturient dairy cows.

Comments 2. NEFA treatment is used at 2.4 mM to model clinical ketosis, but the rationale for this concentration (e.g., based on plasma levels in ketotic cows) is not explicitly stated.

Response 2: We sincerely appreciate this valuable suggestion regarding the NEFA treatment concentration. As recommended, we have now explicitly stated the rationale in the Materials and Methods section (subsection 4.1): The NEFA concentration of 2.4 mM was selected based on documented plasma NEFA levels (2.1-2.6 mM range) in clinically ketotic dairy cows [42], and our dose-response experiments confirmed this concentration effectively induced ketogenic alterations without cytotoxicity. This modification ensures full transparency in the experimental design rationale.

Comments 3. The NTA and TEM data (Fig. 1d, e) show no significant differences between NK and CK EVs, yet the discussion implies higher EV release in diseased states (citing Refs. 25-27). The authors shall quantify EV yield differences statistically, or clarify that no difference was observed.

Response 3: The number of peaks referred to here is shown in Figure 1d and 1e. The peak number of NK source exosomes was 3.5×107 cells/mL, while the peak number of CK source exosomes was 5×107 cells/mL.

Comments 4. SIRT1 overexpression adenovirus construction is detailed, but the multiplicity of infection (MOI) and transfection efficiency are not reported.

Response 4: Thank you for your reminder. After checking the data, it was found that an important standard MOI=100 was missed during the construction of the SIRT1 overexpression adenovirus.

Comments 5. Statistical analysis uses ANOVA, but post-hoc tests (e.g., Tukey's) are not mentioned. The authors need to specify the post-hoc test used and ensure all P-values are from appropriate comparisons.

Response 5: ‌We sincerely appreciate this important suggestion regarding our statistical analysis. As recommended, we have now explicitly stated the post-hoc test details in the Materials and Methods section: Following ANOVA, post-hoc comparisons were performed using Tukey's HSD test for multiple group comparisons, with all P-values derived from appropriate pairwise comparisons. This modification ensures full transparency in our statistical methodology.

Comments 6. Some figures are repetitive (e.g., Figs. 2-7 show similar trends for mRNA, protein, and enzyme activities). Some legends are unclear (e.g., Fig. 2a lacks units for relative expression).

Response 6: The study indeed employed multiple experimental approaches to measure relative mRNA levels and protein expression levels in cell models under different experimental conditions, thereby validating experimental hypotheses and confirming the consistency of results. Notably, many figures present normalized relative values without specifying units, as these measurements are inherently dimensionless by nature.

Comments 7. The optimal EV concentration (40 μg/mL) and time (12 h) are determined via CCK-8, but data show no toxicity at higher doses — why not test functional effects at 60-80 μg/mL?

Response 7: At that time, the experimental process considered that CK source and NK source had obvious differences at 40 μg/mL concentration, and exosome extraction was not easy. Therefore, this concentration was selected for economic and effective consideration.

Comments 8. Overexpression results (Fig. 6-7) show synergistic effects with NK-EVs but antagonism with CK-EVs, why?

Response 8: While adenovirus overexpressing SIRT1 and NK-derived exosomes both upregulated SIRT1 protein and gene expression, CK-derived exosomes significantly suppressed these biological processes. This resulted in a synergistic upregulation and antagonistic downregulation phenomenon under experimental conditions. The study conclusively demonstrates that exosomes modulate hepatic cell lipid toxicity through the SIRT1/SREBP-1c/PGC-1α signaling pathway.

Comments 9. The discussion adequately links findings to the SIRT1 pathway but lacks depth on upstream regulators (e.g., how NEFAs modulate EV cargo) or downstream effects (e.g., mitochondrial function). It also repeats some results without new points. The authors can perhaps expand on potential mechanisms, such as EV-mediated miRNA transfer. In addition, discussion on translational potential should be cautious, suggesting in vivo validation (e.g., EV injection in ketotic cows).

Response 9: Thank you for your valuable suggestions. In our upcoming research, we will conduct in-depth investigations into upstream regulatory factors (e.g., how NEFAs regulate EV cargo) and downstream effects (e.g., mitochondrial function), aiming to provide a more comprehensive explanation for the role of EVs in bovine ketosis. Regarding discrepancies in the results description, as this study employed multiple experimental techniques to explore SIRT1/SREBP-1c/PGC-1α-related indicators, some overlapping descriptions are inevitable. This approach enhances research credibility through multi-dimensional validation. Potential mechanisms such as EV-mediated miRNA transfer will be further investigated and published in subsequent studies.

Minor issues:

Comments 10. The English requires extensive editing for grammar, phrasing, and consistency (e.g., "NE-FAs" vs. "NEFAs"; "hep-EVs" introduced without definition; awkward sentences like "The suggestion that EVs might regulate intercellular communication during disorders of hepatic lipid metabolism has not been addressed in detail"). The manuscript needs to be proofread by a native speaker or professional service.

Response 10: Abbreviations for NEFA and hep-EVs have been added to improve readability for non-specialists. Awkward sentences in the text have been revised to more appropriate expressions.

Comments 11. The abstract is informative but could better highlight key findings (e.g., specific gene/protein changes). Keywords are appropriate but add "ketosis" and "extracellular vesicles" for better indexing.

Response 11: Thank you for your suggestion, which has been reflected in the abstract and keywords.

Comments 12. Introduction section, some citations are outdated or mismatched (e.g., Ref. 13 on NF-κB is from 2015; ensure all align with claims).

Response 12: We sincerely appreciate this valuable feedback regarding our reference quality. As recommended, we have systematically reviewed all citations, replacing outdated references with the latest research to ensure alignment with current scientific consensus. This revision strengthens the methodological foundation of our study.

Comments 13. Gene names in Table 1 should be italicised (e.g., SIRT1). ELISA kits are from "Shanghai Enzyme Link,"—provide full company name or catalogue numbers.

Response 13: The gene names in Table 1 are already italicized, and the full company name and catalogue number of the ELISA kit are added.

Comments 14. Formatting of references is inconsistent (e.g., some journals abbreviated, others not).

Response 14: Thank you. The reference literature format was reviewed and unified.